# Predictive Validity of the Postural Assessment Scale for Stroke (PASS) to Classify the Functionality in Stroke Patients: A Retrospective Study

**DOI:** 10.3390/jcm11133771

**Published:** 2022-06-29

**Authors:** Cecilia Estrada-Barranco, Ismael Sanz-Esteban, Maria José Giménez-Mestre, Roberto Cano-de-la-Cuerda, Francisco Molina-Rueda

**Affiliations:** 1Department of Physiotherapy, Faculty of Sport Sciences, Universidad Europea de Madrid, Villaviciosa de Odón, 28670 Madrid, Spain; cecilia.estrada@universidadeuropea.es (C.E.-B.); ismael.sanz@universidadeuropea.es (I.S.-E.); mariajose.gimenez@universidadeuropea.es (M.J.G.-M.); 2Department of Physical Therapy, Occupational Therapy, Physical Medicine and Rehabilitation, Faculty of Health Sciences, Rey Juan Carlos University, 28922 Madrid, Spain; francisco.molina@urjc.es

**Keywords:** functionality, neurorehabilitation, predictive validity, postural control, stroke, validity

## Abstract

The analysis of the predictive validity of a scale allows us to establish objectives in rehabilitation and to make decisions in the clinical setting. The objective of this study was to determine the validity of the Postural Assessment Scale for Stroke (PASS) to predict functionality at each stage of recovery in stroke patients. Methods: A retrospective study was carried out collecting data from patients admitted to a neurorehabilitation hospital. All patients having suffered a stroke less than two months before hospital admission were included in the study. The balance was measured with the PASS scale and the functionality with the Functional Independence Measure (FIM) scale. Simple linear regressions were performed to model the relationship between the PASS and FIM scores in the acute, subacute and chronic stages (6 and 12 months), as well as between the PASS scores at admission and the FIM values in the chronic stage. Results: The PASS scale showed a good predictive validity (R^2^ values from 0.54 to 0.87; β values from 1.99 to 2.62; *p* < 0.001) for FIM scores at acute, subacute and chronic stages, with lower goodness-of-fit for PASS scores at admission and FIM scores at 12 months (R^2^ = 0.383; β = 1.61 (0.96–2.26); *p* < 0.001). Cut-off points in the PASS scale to predict high functional level were 17.5 for the acute stage and 16.5 for the subacute and chronic stages. A score of 8.5 on the PASS scale measured in the acute phase predicted a high functional level at 12 months. Conclusion: The PASS scale is a useful tool to classify the functionality of stroke patients in the acute, subacute and chronic phases. The PASS score upon admission into the hospital can predict the functionality of the stroke patients after 12 months. However, future studies should be carried out to corroborate our findings with larger sample sizes.

## 1. Introduction

Stroke is one of the leading causes of disability in adults worldwide [1,2]. In the European Union, stroke constitutes the first cause of death, and one of the main causes of adult disability [3]. Every year approximately 1.1 million inhabitants are affected by a stroke, and it causes 440,000 deaths per year [4,5]. Nevertheless, the survival rate in stroke events is rising, and it is expected to increase over the next few years [6].

Postural control is usually altered after a stroke, which constitutes major social, economic and personal burdens of this population [7,8]. Therefore, to assess postural control is a key element in a rehabilitation context [9,10]. The relationship between postural control and functionality at each moment could guide rehabilitation and help establish functional goals in each stage. The Functional Independence Measure (FIM) has been considered the gold standard in the evaluation of functionality in stroke patients [11]. It provides more accurate information than what is obtained with the Barthel Index (BI). However, it requires prior knowledge of the patient, the observation of the functional development of each task assessed, the evaluation sometimes takes several days/sessions and it is necessary to know and follow a decision flow established in the instructions of the scale [12].

In this context, the analysis of the predictive validity of a scale could allow us to know the ability of that scale to predict the score of another scale or of another event studied. In addition, establishing the predictive validity through a simpler scale allows us to obtain information more quickly, which facilitates its clinical application. To establish objectives in rehabilitation and to make decisions in the clinical setting, it is especially important to have predictive parameters that allow us to anticipate certain events and establish priorities in rehabilitation. In the study of predictive validity in stroke, trunk control and balance have been shown to be good indicators of motor recovery in patients. Postural control is essential for many activities such as breathing, swallowing and all activities of daily living [13]. Different researchers have studied the predictive capacity of the trunk control, studying other constructs like the risk of falls, the ability to walk or dependency in stroke patients [14,15,16,17]. However, to our best knowledge, the predictive validity of balance and postural control to predict functionality in the different stages of recovery after stroke has not been studied.

The Postural Assessment Scale for Stroke (PASS) measures balance in lying down, sitting and standing positions in stroke patients. It consists of 12 items whose score can vary from zero to three, with zero being the lowest level of functionality and three being the highest level. The total score can be 36 [18]. It was developed taking into account the relationship between the ability to maintain a posture and to ensure balance when changing position. PASS scale is sensitive to detect changes even in patients with great postural deficit and has an increasing difficulty in the scale itself that allows its application in patients of different functional levels [19]. In addition, the PASS scale has been shown to be more sensitive than the Berg Balance Scale (BBS) and the balance subscale of the Fugl–Meyer scale (FM-B) in patients with severe stroke in early stages [20], as well as for the assessment of balance in patients with more severe affectations [21]. It has also been shown to be useful in quantifying progress in patients with pusher syndrome [22].

However, to our best knowledge, there are no previous works that have studied the capacity of the PASS scale in predicting functionality in patients with acute, subacute and chronic stroke. Therefore, the objectives of this study were: (1) to analyze the predictive validity of the PASS scale to establish the functionality at each stage of recovery after stroke; and (2) to determine if the PASS score upon admission at the hospital could predict the functionality of the stroke patients after 12 months.

## 2. Materials and Methods

### 2.1. Design

A retrospective study was carried out collecting data from patients admitted to Hospital Los Madroños (Madrid), a neurorehabilitation hospital in Madrid, Spain. STROBE (STrengthening the Reporting of OBservational studies in Epidemiology) guidelines were followed to standardize the reporting of this work. The study was approved by the Local Ethics Committee (031020168316).

### 2.2. Participants

Patient’s data were collected for study purposes from medical records stored in the hospital database (CE-B). The screened period was from October 2016 to May 2017. All patients having suffered a stroke less than 8 weeks before hospital admission were included in the study if: (1) gait and balance treatment had been identified as a goal by the patient and the rehabilitation team, according to SMART goals [23]; (2) stroke was confirmed by a neurologist using either magnetic resonance imaging or computed axial tomography; and (3) all patients should follow the same individual rehabilitation treatment oriented to gait and balance improvements. Those patients with clinical situations unrelated to the stroke that could interfere with their recovery (skeletal muscle pathologies, unstable cardiovascular or neurological situation) were excluded.

### 2.3. Procedure

The PASS scale was used to measure balance, and the Functional Independence Measure (FIM) was selected to assess the functionality of the patients. Scores of both scales were collected at 4 time points during the rehabilitation process: upon admission, at 3 months (subacute state), at 6 months and at 12 months (chronic state) [24].

The PASS is a balance test. This measure evaluates static balance, in sitting and standing, and dynamic, in position changes including the lying position. It consists of 12 items. Each item is evaluated from 0 to 3, where 0 is the lowest functional level and 3 is the highest functional level. The total score is 36 points [19]. The Spanish version of the PASS was administered [25].

The FIM scale evaluates different aspects of functionality such as self-care, sphincter control, locomotion or mobility. It consists of 18 items, 13 of which evaluate physical function and are related to the Barthle index, and 5 additional items which evaluate cognitive function in aspects such as problem execution and memory, in addition to social interaction [26]. Each item is evaluated in 7 categories (1 is completely independent, 7 is completely independent) [27]. Thus, the lowest score on the scale is 18 and the highest is 126. The FIM scale has shown high sensitivity in general, especially regarding the level of attention required for functional activity [28]. Specifically, it has been shown to be robust, in terms of its psychometric properties, in hospitalized stroke patients [29]. Different studies have established cut-off points to establish categories with respect to functionality using the FIM scale [28,29]. Inouye et al. categorized patients into three groups according to their FIM score (≤36; 37–72; ≥73), corresponding to severe, moderate and mild impairment of functionality, respectively [30].

The psychometric properties of the administered scales (PASS and FIM) have been studied, showing that they are valid and reliable for evaluating their respective constructs in stroke patients [27,31,32].

### 2.4. Sample Size

The sample size was calculated using the G*Power software (version G*Power 3.1.9.2, Universität Düsseldorf, Düsseldorf, Alemania). The following parameters were considered: bilateral contrast, an effect size of 0.15 (medium), an error alfa of 0.05, a power of 80% and number of predictors (*n* = 1). The resulting sample size was 55 participants.

### 2.5. Statistical Analysis

#### 2.5.1. Predictive Models

A single linear regression was performed to model the relationship between the PASS and FIM scores. A model was run for each of the studied time points (acute, subacute and chronic states) and between PASS scores at admission and FIM values in the chronic stage. A 95% confidence interval was established for the β coefficient. The validity of the model was analyzed by calculating the ANOVA, with significance set at *p* < 0.001. The Durbin and Watson test, and the Kolmogorov–Smirnov test to analyze the normality of residuals in all models, were used. 

#### 2.5.2. Cut-Off Points

Receiver operating characteristic (ROC) curves were used to determine the discriminatory power of the PASS for classifying participants into 2 groups based on their FIM score: severe and moderate functional impairment (FIM < 73) and mild functional impairment (FIM ≥ 73) [30].

The accuracy was assessed using the area under the curve (AUC), which can be interpreted as the probability of correctly identifying participants with minor or moderate risk of falls. An AUC value of 0.9 and above indicates high accuracy, 0.7 to 0.9 indicates moderate accuracy, 0.5 to 0.7 indicates low accuracy and 0.5 and below indicates test due to chance [33,34]. The cut-off point of a continuous variable that determines the highest sensitivity and specificity is the value which has the highest Youden index, calculated according to the expression: sensitivity + specificity − 1 [35].

## 3. Results

Data from 375 patients were retrospectively analyzed. Up to 61 patients (40 males, 21 females; mean age 62.75 ± 13.31 years) met the eligibility criteria and were included. Of them, 47 (77.0%) presented ischemic stroke. Data from the 61 patients were upon admission (17.07 ± 13.21 days) and the subacute state (107.06 ± 13.25 days), from 58 patients at 6 months (197.08 ± 13.25 days; chronic state) and from 42 patients at 12 months (382.06 ± 9.8 days; chronic state). The main reason of patients lost was the discharge from the hospital. The characteristics of participants are presented in Table 1.

### 3.1. Predictive Models

Significant relationships were found between PASS scores and FIM scores at each stage of stroke recovery, and between PASS scores upon hospital admission and FIM scores at 12 months. Table 2 shows the parameters of the linear regression models. Both the variability in the target variable which was explained by the regression model and the increase shown by the β coefficient increased when advancing through the sequential stages defined in the rehabilitation process. On the contrary, lower goodness-of-fit was found for PASS scores at admission and FIM scores at 12 months.

### 3.2. Cut-Off Points

AUC values were 0.828 in the acute stage, 0.877 in the subacute stage, 0.901 in the chronic stage (6 months) and 0.966 in the chronic stage (12 months). In all cases, the AUC was greater than 0.7 and in the chronic stage was greater than 0.9, which indicates that the PASS may be valid to identify stroke patients with mild functional impairment measured with the FIM scale (Figure 1).

The cut-off point was calculated to establish the level of functionality associated with the PASS scale scores through the Youden Index. The PASS cut-off points were 17.5 for the acute stage and 16.5 for the subacute and chronic stages. Therefore, a value above these scores on the PASS scale would be related to mild functional impairment.

The PASS scale upon hospital admission was also shown to be a valid instrument to predict functionality using the FIM scale at 12 months (AUC = 0.832). The PASS cut-off was 8.5 which indicates that values above this score may predict a high level of functionality at 12 months (Figure 2).

## 4. Discussion

The results obtained in this work show that PASS was a useful tool to predict the functionality of stroke patients in the acute, subacute and chronic phases. In addition, the present study established PASS cut-off values predicting mild functional impairment (17.5 at the acute stage and 16.5 at the subacute and chronic stages). 

Our results also showed that PASS assessment in the acute phase was a useful tool to predict the functionality of stroke patients at 12 months, establishing that PASS scores above 8.5 points upon hospital admission were able to identify a higher functional level at 12 months in stroke patients. It must be noticed that a FIM score ≥ 73 was considered as a high level of functionality [28]. In this line, another study found that a better FIM score (score ≥ 73) was correlated with a lower risk of falls and functional deterioration [36], so future studies should be conducted to predict these constructs through postural control scales.

The validation of predictive models has been widely studied in rehabilitation. Several factors in the acute phase, such as advanced age, extent of the injury, presence of dysphagia, stroke complications (cerebral edema), absence of sphincter control and paresis of the upper limbs, have been identified to be related with worse functional recovery after stroke [37,38,39]. Trunk control has also been noted as a predictor of walking ability at 12 months and of reduction in the risk of falls [40]. The ability of different trunk control scales to predict functionality has also been studied. However, none of the scales studied has been shown to be complete, in absolute terms, to predict functionality at each stage of recovery and 12 months after the stroke [41].

A predictive validity makes it possible to reduce the number of evaluations necessary to assess the patient’s global condition, allowing to set individualized rehabilitation objectives and good relationships between the two constructs studied [32]. Additionally, postural control has shown to be a predictor of important aspects of recovery, such as walking ability or the risk of falls [42]. In this context, the PASS scale is a scale originally developed for the evaluation of stroke patients. It evaluates both static and dynamic balance and analyzes postural control in basic activities such as postural changes in lying, sitting and standing. In addition, it has shown to be sensitive and effective in highly affected patients and in patients with a high functional level [20]. Our results confirm the predictive relationship that exists between postural control and functionality in the different stages of recovery after a stroke. 

Previous studies had analyzed the validity of the PASS scale in acute patients in predicting functionality in the chronic stage. Hsieh et al. [43] analyzed the effect of different factors such as age and functionality in the acute phase (measured with the FIM scale or PASS score), on the ability to perform daily living activities 6 months after stroke, and determined that the PASS score was the main predictive factor. Our results are in accordance with those results and extended this relationship up to 12 months. 

Huang et al. [44] also analyzed the ability of the PASS scale to predict walking ability in stroke patients, defining a cut-off value of 12.5 upon admission as a predictor of the ability to walk, with or without technical assistance at discharge (after a mean hospital stay of 18.12 days). In our study, PASS cut-off values in the acute phase predicting a high level of functionality in the subacute and chronic phases were 17.5 and 16.5, respectively. Differences could be interpreted as that greater trunk control and balance are required when attempting to predict global functionality beyond just ambulation. In the present work, the FIM scale was used to evaluate functionality, a scale in which only two of the total eighteen items evaluate locomotion. Although ambulation and functional level are closely related, other competences could be necessary to achieve a higher functional level.

This work presents several limitations. The calculated sample size required 55 participants and, because of discharges to the rehabilitation center, the sample evaluated at 12 months was 42 patients. In addition, the sample size was not large enough to analyze the data according to the type or severity of the stroke, as well as the location of the lesion. Future studies are warranted to corroborate the PASS cut-off values for predicting functionality in patients with stroke in its different described phases. Although this work provides a direct and quantified relationship between postural control and functionality, it would be interesting to be able to establish a multivariate predictive model with larger sample sizes.

## 5. Conclusions

The PASS scale was a useful tool to predict the functionality of stroke patients in the acute, subacute and chronic phases. The PASS cut-off values to determine a high functional level were 17.5 at the acute stage and 16.5 at the subacute and chronic stages. 

The PASS score upon admission in the hospital can predict the functional impairment of the stroke patients after 12 months. We established that a score above 8.5 points in the PASS scale upon admission in the hospital is a cut-off point to identify a higher functional level in the stroke patients at 12 months. A score of 8.5 on the PASS scale measured in the acute phase predicted a high functional level at 12 months. However, future studies should be carried out to corroborate our findings with larger sample sizes.

## Figures and Tables

**Figure 1 jcm-11-03771-f001:**
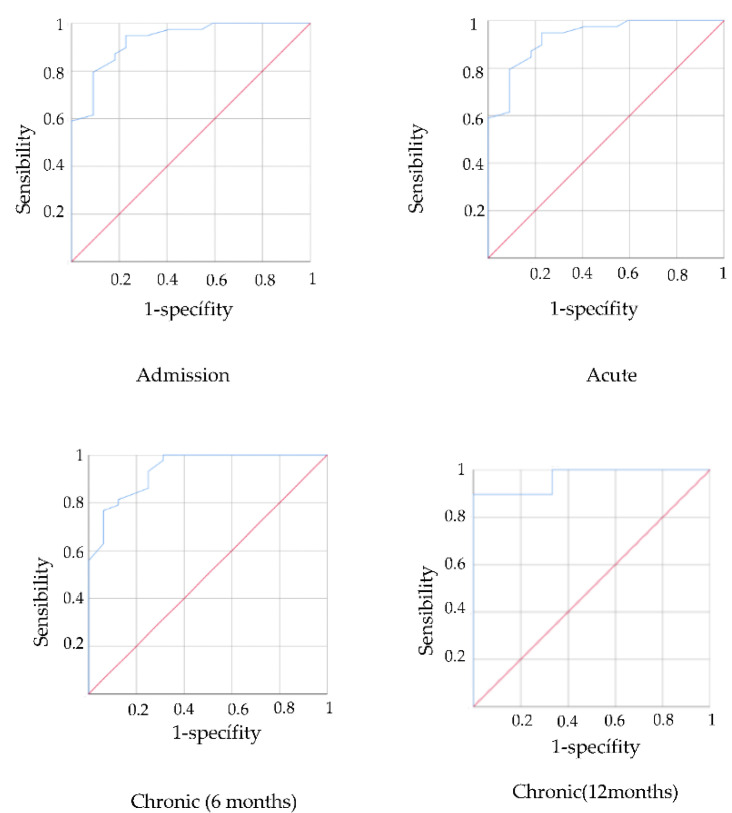
ROC curves predictive validity PASS for functionality in acute, subacute and chronic stage.

**Figure 2 jcm-11-03771-f002:**
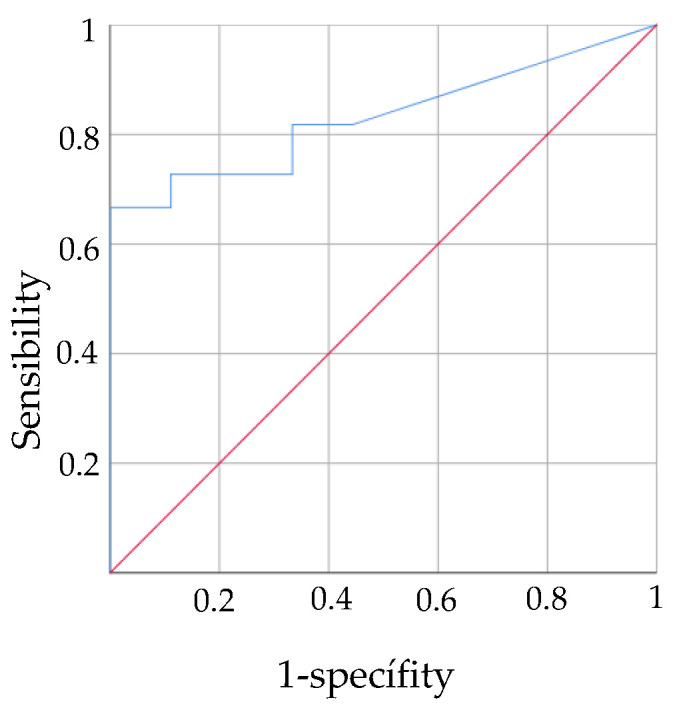
ROC curves predictive validity PASS in acute stage for functionality in chronic stage.

**Table 1 jcm-11-03771-t001:** Characteristics of participants.

Variable	Acute/ Subacute Stage	Chronic Stage (6 Months)	Chronic Stage (12 Months)
*n*	61	58	42
Age, y	62.75 (±13.31)	65.76 (±13.48)	62.42 (±12.4)
Sex (male/female), *n*	40/21	37/21	29/13
Type of stroke (ischemic/hemorrhagic), *n*	47/14	44/14	31/11

**Table 2 jcm-11-03771-t002:** Linear Regression Results (PASS–FIM).

Model	R^2^	ANOVA	β	*p*	Durbin	K-S ^1^
PASS ^2^ 0–FIM ^2^ 0	0.540	<0.001	1.99 (1.52–2.48)	<0.001	2.150	0.200
PASS 3–FIM 3	0.658	<0.001	2.15 (1.75–2.56)	<0.001	2.374	0.060
PASS 6–FIM 6	0.729	<0.001	2.48 (2.078–2.88)	<0.001	1.966	0.162
PASS 12–FIM 12	0.867	<0.001	2.62 (2.29–2.95)	<0.001	1.672	0.082
PASS 0–FIM 12	0.383	<0.001	1.61 (0.96–2.26)	<0.001	1.858	0.141

^1^ K-S: Kolmogorov Smirnov; ^2^ PASS: Postural Assessment Scale for Stroke Patients; ^3^ FIM: Functional Independence Measure.

## Data Availability

No new data were created or analyzed in this study.

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
