# Peer review of "Predictive Validity of the Postural Assessment Scale for Stroke (PASS) to Classify the Functionality in Stroke Patients: A Retrospective Study"

_jcm, 2022, doi:10.3390/jcm11133771_

Round 1

Reviewer 1 Report

Estrada-Barranco et al tried to evaluate whether PASS scale could predict functionality of patients with stroke at different stage. Despite the idea was interesting, there were major several issues must be addressed.

1. I noticed that the correlation between PASS 0  and FIM 12 was very weak, which undermine the clinical value of your results. Could FIM at discharge predict itself at follow-up (in your data set or published articles)?  If so,  there is no need to use PASS to indirectly predict FIM at follow-up, which would fundmentally weaken the novelty of your manuscript.

2. Usually, auhtors of research articles would summarize the baseline characteristics of enrolled patients in Table 1(like age, sex, treatment strategy), which would help the readers understand the context in which conclusions are made. Therefore, I suggest you providing this information.

3.Univariate linear regression is equal to Pearson or Spearman correlation analysis. I suggest you using multivariate linear regression to control important confounders (like age, BMI, disease severity, treatment strategy). 

4. In a study with small simple size, the results of ROC analysis are unstable. The generalizability of the cutoff value in the present study is a big question.

Author Response

Response letter manuscript

jcm-1743234

Predictive validity of the Postural Assessment Scale for Stroke (PASS) to classify the functionality in stroke patients: a retrospective study

We would like to thank the editor and reviewers for their comments in this review, which have greatly improved the readability of the manuscript. We would like to inform you that we have edited the manuscript according to the very constructive suggestions from the reviewers.

Below, please find a list of revisions and a response to each of the reviewer’s comments. We have highlighted all changes in the manuscript in yellow. We hope that the revisions in the manuscript and our accompanying responses will be enough to make our manuscript suitable for publication in the Journal of Clinical Medicine

We shall look forward to hearing from you at your earliest convenience.

Yours sincerely,

The authors

Reviewer 1

I noticed that the correlation between PASS 0 and FIM 12 was very weak, which undermine the clinical value of your results. Could FIM at discharge predict itself at follow-up (in your data set or published articles)?  If so, there is no need to use PASS to indirectly predict FIM at follow-up, which would fundamentally weaken the novelty of your manuscript.

We appreciate the reviewer's comment. The goal of this analysis was to predict the functionality score using a scale that assesses trunk control and balance. We consider that this prediction could reduce the time of evaluation in a clinical scenario since the PASS scale is an easier and faster instrument to administer to patients, which does not require training. The FIM is a long and a complicated scale that requires training.

We have reinforced this aspect in the introduction section by including two bibliographical references.

In the second paragraph we add:

The Functional Independence Measure (FIM) has been considered the gold standard in the evaluation of functionality in stroke patients [11]. It provides more accurate information than what is obtained with the Barthel Index (BI). However, it requires prior knowledge of the patient, the observation of the functional development of each task assessed, the evaluation sometimes takes several days/sessions and it is necessary to know and follow a decision flow established in the instructions of the scale [12]

“In addition, establishing the predictive validity through a simpler scale allows obtaining information more quickly which facilitates its clinical application”.

Usually, authors of research articles would summarize the baseline characteristics of enrolled patients in Table 1(like age, sex, treatment strategy), which would help the readers understand the context in which conclusions are made. Therefore, I suggest you providing this information.

We appreciate your suggestion.

We have included table 1 in the document with the characteristics of the sample

3.Univariate linear regression is equal to Pearson or Spearman correlation analysis. I suggest you using multivariate linear regression to control important confounders (like age, BMI, disease severity, treatment strategy).

We appreciate your suggestion. The referee is right, both type of tests analyzes the relation between two variables. However, the slight difference is that while correlation quantifies how they are related (linear, non-linear), the result of linear regression is an equation (model) that could help us in predicting the value of one variable (FIM in our study) considering the value of the other (PASS in our study). In addition, multivariate linear regression requires a larger sample to ensure that the different variables are representative. In our case, the sample is limited.

However, following your suggestion, we have performed a multimodal regression including the descriptive data of the sample and the PASS and FIM data at baseline. The model is significant for age, type of stroke, and PASS at baseline. Not so for sex or for the FIM. However, since the number of patients with ischemic stroke is quite higher than that ones with hemorrhagic stroke in our sample (47 ischemic, 14 hemorrhagic), we consider that our data are not valid to consolidate this model.

Nevertheless, we consider that our findings allow us to provide a predictive relationship between trunk control and functionality, although it is true that additional studies with a larger sample are necessary to confirm these findings.

Only for the reviewer´s knowledge, the results of the multimodal regression are shown in the following table:

MODEL

β

  P  

R2

adjusted

ANOVA

Durbin

K-S1

0.534

<0.001

1.86

0.2

FIM 0

0.2   (-0.20-0.59)

0.33

Type of stroke (ischemic/

hemorrhagic)

0.29  (5.37-37.54)

0.01

Age

-0.24 (-1.25-0.01)

0.048

PASS 0

0.43  (0.1-2.15)

0.03

We do think that our Results could answer to the aim of our study. Maybe our sample size would not be the adequate to conduct a multimodal regression to answer our objectives.

  1. In a study with small simple size, the results of ROC analysis are unstable. The generalizability of the cutoff value in the present study is a big question.

The reviewer is right, the text has been re-written in relation to this issue.

Reviewer 2 Report

interesting work well written, but with numerous weaknesses:

1-small simple size

2-retrospective study

by admission of the authors the purpose is to evaluate the PASS score, but the FIM score is introduced and not mentioned in the discussion, which should be improved, explaining why the use the score and discussing the results they obtained in this way

Author Response

Response letter manuscript

jcm-1743234

Predictive validity of the Postural Assessment Scale for Stroke (PASS) to classify the functionality in stroke patients: a retrospective study

We would like to thank the editor and reviewers for their comments in this review, which have greatly improved the readability of the manuscript. We would like to inform you that we have edited the manuscript according to the very constructive suggestions from the reviewers.

Below, please find a list of revisions and a response to each of the reviewer’s comments. We have highlighted all changes in the manuscript in yellow. We hope that the revisions in the manuscript and our accompanying responses will be enough to make our manuscript suitable for publication in the Journal of Clinical Medicine

We shall look forward to hearing from you at your earliest convenience.

Yours sincerely,

The authors

Reviewer 2

Interesting work well written, but with numerous weaknesses:

1-small simple size

Thank you very much for all your comments.

We agree with the reviewer, the sample is limited. We have incorporated this issue in the limitation section and conclusions. Studies with more sample are needed in this field.

At the end of the discussion, we have included:

“Although this work provides a direct and quantified relationship between postural control and functionality, it would be interesting to be able to establish a multivariate predictive model for which a larger sample size.”

And also, in the conclusions we have incorporated:

“The PASS score upon admission in the hospital can predict the functional impairment of the stroke patients after 12 months. We established that a score above than 8.5 points in the PASS scale upon admission in the hospital is a cut-off point to identify a higher functional level in the stroke patients at 12 months. A score of 8.5 on the PASS scale measured in the acute phase predicted a high functional level at 12 months. However, studies with a larger sample should be carried out to corroborate our findings with larger sample sizes.”

The conclusions have also been modified in the abstract:

“Conclusion: The PASS scale is a useful tool to classify the functionality of stroke patients in the acute, subacute, and chronic phase. The PASS score upon admission in the hospital can predict the functionality of the stroke patients after 12 months. However, studies with a larger sample should be carried out to corroborate our findings with larger sample sizes.”

2-retrospective study

By admission of the authors the purpose is to evaluate the PASS score, but the FIM score is introduced and not mentioned in the discussion, which should be improved, explaining why the use the score, and discussing the results they obtained in this way

Following the reviewer's suggestion, we have included in the Discussion this sentences and its bibliographic references.

“It must be noticed that a FIM score ≥73 was considered as a high level of functionality [29]. In this line, another study found that a better FIM score (score ≥73) was correlated with a lower risk of falls and functional deterioration [37], so future studies should be conducted to generated to predict these constructs through postural control scales.”

Round 2

Reviewer 1 Report

The authors's revison adressed most of my concerns. But please check whether Fig.1 Chronic 12 months is appropriate. It seems the AUC is 100%?

Author Response

Response letter manuscript

jcm-1743234

Predictive validity of the Postural Assessment Scale for Stroke (PASS) to classify the functionality in stroke patients: a retrospective study

We would like to thank the editor and reviewers for their comments in this review, which have greatly improved the readability of the manuscript. We would like to inform you that we have edited the manuscript according to the very constructive suggestions from the reviewers.

Below, please find a list of revisions and a response to each of the reviewer’s comments. We have highlighted all changes in the manuscript in yellow. We hope that the revisions in the manuscript and our accompanying responses will be enough to make our manuscript suitable for publication in the Journal of Clinical Medicine

We shall look forward to hearing from you at your earliest convenience.

Yours sincerely,

The authors

Reviewer 1

The authors's revison adressed most of my concerns. But please check whether Fig.1 Chronic 12 months is appropriate. It seems the AUC is 100%?

We would like to thank the reviewer for their positive comments in this review, which have greatly improved the readability of the manuscript. We would like to inform you that we have edited the manuscript according to the very constructive suggestions.

Thank you very much for this observation. You are totally right. It was a mistake. We have reviewed all figures and changed figure 2 according to the results:

Table 2: Linear Regression Results (PASS-FIM)

MODEL

R2

ANOVA

β

  P  

Durbin

K-S1

PASS2 0–FIM2 0

0.540

<0.001

1.99 (1.52-2.48)

<0.001

2.150

0.200

PASS 3-FIM 3

0.658

<0.001

2.15 (1.75-2.56)

<0.001

2.374

0.060

PASS 6–FIM 6

0.729

<0.001

2.48 (2.078-2.88)

<0.001

1.966

0.162

PASS 12–FIM 12

0.867

<0.001

2.62 (2.29-2.95)

<0.001

1.672

0.082

PASS 0–FIM 12

0.383

<0.001

1.61 (0.96-2.26)

<0.001

1.858

0.141

1K-S: Kolmogorov Smirnov; 2PASS: Postural Assessment Scale for Stroke Patients; 3FIM: Functional independents measure.

Figures

Figure 1: ROC curves predictive validity PASS for functionality in acute, subacute and chronic stage (see word)

Figure 2: ROC curves predictive validity PASS in acute stage for functionality in chronic stage (see word)
